# The mosaic of science: Disciplinary diversity and scientific prestige in research groups in Colombia

Julián D. Cortés  *

School of Management and Business, Universidad del Rosario, Bogotá, Colombia

* julian.cortess@urosario.edu.co

## Abstract

Collaboration among science teams is essential for addressing complex global challenges. A key feature of such collaboration is disciplinary diversity; however, its relationship with team performance remains debated. Existing research has focused primarily on high-income countries and has relied on proprietary databases, often overlooking the distinctive scientific ecosystems of middle- and low-income nations. This geographical and methodological bias has created a gap in understanding how team composition affects scientific outcomes in these underrepresented contexts. This study examines a ten-year period using publicly available data from all Colombian research groups maintained by the Ministry of Science, Technology, and Innovation (MinCiencias). Disciplinary diversity was measured using the *DIV* indicator proposed by Leydesdorff et al. We show that the relationship between disciplinary diversity and scientific prestige is non-linear and moderated by both group size and broad disciplinary area. Our analysis identifies two main findings: low diversity consistently characterizes research groups with a declining performance trajectory, and groups that advance in national rank exhibit a statistically similar diversity structure to those following a volatile trajectory in the national ranking. These results challenge the assumption that increasing diversity necessarily leads to better performance. Instead, they indicate that the functional role of diversity is not monotonic and that an optimal, context-specific level may exist. This nationwide study contributes to science policy by demonstrating that fostering field-dependent diversity structures, rather than maximizing diversity indiscriminately, may be critical for strengthening integrative and transformative research systems in emerging economies.

## 1 Introduction

Modern society and science face complex challenges, such as climate change, pandemics, or extreme inequality. Overcoming these difficulties will require a concerted, collective effort. Scientific collaboration emerges as a rich tapestry woven

**Data availability statement:** All data underlying the findings of this study are publicly available from the Ministry of Science, Technology and Innovation of Colombia. National Groups Assessments data are available at https://www.datos.gov.co/d/hrhc-c4wu, and National Researchers Assessment data are available at https://www.datos.gov.co/d/bqtm-4y2h.

**Funding:** The author(s) received no specific funding for this work.

**Competing interests:** The authors have declared that no competing interests exist.

with diverse threads of thought contributed by individuals from different institutions, nations, and areas of expertise [1–3]. A key aspect of intricate modern scientific collaboration is the aforementioned disciplinary diversity, which entails examining fundamental questions regarding team composition, for example: What is the level of cognitive diversity within science teams? How can it be measured? Is there a relationship between diversity and the scientific team's performance, such as scientific prestige and trajectory? [4–9].

These questions have fostered extensive research that examines teams in multiple organizational contexts. Of greater significance is analyzing such diversity and its relationship with team effectiveness and performance [10–14]. Particularly, research in the field of the science of team science (SciTS) has examined the composition and dynamics of science teams, the role of different disciplines in their effectiveness, as well as how science teams are measured and assessed [15].

The concentration of studies in high-income countries represents a limitation of the existing literature. This geographical focus is reinforced by the reliance of quantitative research methodologies (e.g., bibliometric analysis applied at the journal, document level) on canonical bibliographic databases, such as Scopus or Web of Science, which may not provide comprehensive coverage of scholarly work from middle- and low-income countries. This research agenda is limited because these databases underrepresent the research landscape of middle- and low-income countries, which face distinct challenges in their science systems compared with those faced by higher-income countries [16–18]. Furthermore, a stream of SciTS seeks to examine the capacity of science teams to advance innovation and generate specific scientific and policy outcomes, including how various dimensions—such as geographic, cognitive, social, organizational, and institutional proximity—shape these results [19]. An examination of these science team dynamics, however, has typically fallen outside the scope of most evaluations [20].

Therefore, this study aims to quantify the disciplinary diversity and analyze its relationship with scientific prestige and trajectory across all research groups in Colombia, using official public government open access data. To achieve this aim, this study makes four contributions to the existing literature. First, by enhancing the understanding of diversity within science teams and research groups and their performance, particularly within the context of science systems in middle and low-income countries. Second, by offering a thorough national evaluation across every field, including those often excluded, such as the social sciences and humanities. Third, this study provides a systematic and longitudinal assessment of science teams and research groups' disciplinary diversity composition, which extends beyond descriptive and case-study-based insights, thereby addressing the needs identified in recent consensus reports from the SciTS community reflecting the view that diversity is a process rather than a static state [21]. Fourth, by incorporating a broader range of data through the use of national-scale, open-access datasets, providing an official and comprehensive evaluation of research groups' disciplinary diversity as well as their scientific reputation, which also highlights the importance of an in-depth understanding of this phenomenon at the national-individual level [22,23]. In sum, this

study aligns with the organizational approach, which focuses on the combination of researchers' organizational affiliations and professional qualifications [24].

The following section will present a scoping review of the science of science literature that focuses on the topic of diversity. Then, we will describe the methodology: methods, data source, sampling, and variables. Subsequently, the results section quantifies this disciplinary diversity and analyzes its association with scientific prestige and trajectory. Finally, the discussion contrasts these findings with the relevant literature, and the paper concludes by summarizing the main findings, acknowledging the study's limitations, and proposing a future research agenda.

## 2 Related literature

Fulfilling the first step of the aim of this research (i.e., the quantification of disciplinary diversity) requires the adoption of a robust conceptual framework. Disciplinary diversity, in the context of teams, can be defined as the variation among team members' skills, abilities, and expertise [25]. The debate concerning the measurement of disciplinary diversity arises from three fundamental issues: the lack of consensus on its definitions, inconsistent results observed across different datasets, and the intrinsic complexity that characterizes contemporary scientific research [26–28]. Despite this, there is a shared view that diversity can be framed by examining three essential features [29–31]: *variety, balance,* and *disparity*. These properties are observable in science teams and research groups [12,32].

*Variety* (i.e., the number of categories present) in the contexts of science teams and research groups, indicates the number of distinct disciplines within a broader field, such as sociology, biochemistry, or physics, in which its members are proficient. *Balance* (i.e., the distribution of elements across *variety* categories), refers to the equitable representation of these disciplines. For example, a research group comprising equal proportions of researchers from sociology, biochemistry, or physics, would exhibit greater balance than a group where economists make up 95% of the personnel. And third, *disparity* (i.e., the differences among *variety* categories), highlights the intellectual, cognitive, or methodological distance separating fields. Hence, a greater difference in those features results in higher disparity. To illustrate, a research group project involving econometricians, leveraging quantitative statistical models, and anthropologists, utilizing qualitative ethnographic methods, will show a greater disparity compared to a collaboration between researchers in organizational studies and applied psychology, both sub-groups proficient in using qualitative methodologies.

Among the proposals to quantify diversity in scholarly communication, teams, and research groups, the *DIV* indicator offers a robust approach [33,34]. This indicator expands upon the Rao-Stirling diversity index [30] introducing methodological refinements. A fundamental characteristic of *DIV* is its independent evaluation of the three constituent components of diversity—variety, balance, and disparity—prior to their integration. This methodological separation ensures that the indicator satisfies the principle of monotonicity, whereby an increase in any single component elevates the overall diversity value, provided the other two remain constant [35]. Consequently, *DIV* enhances interpretability and resolves the 'dual concept' issue by avoiding the premature conflation of variety and balance. Furthermore, its validity is supported by a correlation with established measures of multidisciplinarity, such as the betweenness centrality index, and its adherence to the structural and validation requisites of interdisciplinarity research [26,36].

The *DIV* indicator has been more recently applied in studies concerning interdisciplinary research (IDR). For instance, it has been used to analyze the IDR of research portfolios in Chinese universities [37] or the degree of IDR in social sciences [38]. Other applications include measuring novel approaches to interdisciplinarity, evaluating scientific collaboration, and assessing the scientific impact and social media attention of IDR [39–42]. Additionally, *DIV* has been employed to explore the complexity introduced when considering the time window in relation to the impact of IDR [43]. Notwithstanding this progress, the application of *DIV* to assess disciplinary diversity within science teams and research groups remains to be explored —with few exceptions [32].

Beyond addressing this measurement gap, the second component of the aim of this study (i.e., exploring the potential relationship between disciplinary diversity and team performance and trajectory) is an inquiry on the assumption that such

diversity is inherently functional for team effectiveness. As a result, innovation, impact, and external visibility will manifest organically. Achieving effective disciplinary diversity is, in reality, a complex undertaking. These include the deep knowledge integration required among diverse members, large team sizes, potential goal misalignment, the permeability of membership boundaries, geographic dispersal, and high task interdependence, among others [20]. Wallrich et al.'s [10] meta-analysis' on the relationship between team diversity (i.e., cognitive, job-related, contextual) and performance based on a sample of 615 studies, found that diversity explained less than 1% of the variance in performance. Yet, contextual factors were identified as key moderators, with the diversity-performance link proving more positive for teams engaged in knowledge intensive tasks (e.g., creative or R&D related activities) where a wider range of perspectives and skill sets is beneficial. A comprehensive study indicates that academic diversity within scientific teams has increased, as demonstrated by metrics such as academic entropy, standard deviation, and disparity, particularly in STEM (Science Technology Engineering and Mathematics) fields and high-income countries. Despite this, higher levels of academic diversity do not have a clear association with teams that produce disruptive innovations at higher rates [44]. In sum, evidence showed that, while diversity may offer benefits in specific, cognitively demanding contexts, it is not a universal performance enhancer.

Taken together, this approach will give us a thorough nationwide analysis of research groups' diversity and track their diversity over time, taking into account the field of study, and investigating how it relates to scientific reputation and trajectory, making it reproducible and comparable over time in the process. Moreover —notwithstanding the difficulties and limitations— this study does not concentrate on the degree of interdisciplinarity in research outputs (a primary focus in the existing literature), but rather the cognitive integration processes occurring during the research among researchers [34], an understudied aspect of individuals and team members [45].

## 3 Methodology

### Methods

This study aims to quantify the disciplinary diversity and analyze its relationship with scientific prestige and trajectory across all research groups in Colombia. We use the national classification system, which evaluates research groups across five hierarchical ranks—A1, A, B, C, and *Reconocido*, where A1 groups represent those with the highest prestige— based on criteria defined by MinCiencias, which include research output, collaboration, and contributions to researcher training and innovation. This variable is explained in detail in Section 3.3.2.

In the first stage of our presentation of Results (Section 4) we used the median of research group *DIV* to illustrate temporal evolution, and to compare groups controlling for six major disciplinary areas, national rank, and research group size. Then, in the second stage, we applied a Kruskal-Wallis H-test to compare the median *DIV* across the four national rank trajectory groups: *advancement, decline, stagnation,* and *volatile* (Section 3.3.3). A Dunn's post-hoc test with a Bonferroni correction was applied for pairwise comparisons following the Kruskal-Wallis test results. We used the R statistical environment for analysis [46], along with broadly known packages for data wrangling such as data.table and dplyr [47,48], as well as GWalkR for data visualization [49].

### Data

The data for this study were sourced from the open-access datasets curated by the Colombian Ministry of Science, Technology, and Innovation: MinCiencias [47]. The data are the result of a national endeavor to survey information on researchers, research groups, and institutions, intended to strengthen the National System of Science, Technology, and Innovation. This evaluation framework has evolved since the early 1990s through the implementation of successive calls and assessments [50]. The datasets used in this study contain information from national assessments conducted in 2013, 2014, 2015, 2017, 2019, and 2021. There were no national assessments conducted by MinCiencias in 2016, 2018, and 2020. This analysis incorporates two specific datasets: one detailing research groups and another containing research results. The links to both datasets are shared below:

- National Groups Assessments: https://www.datos.gov.co/d/hrhc-c4wu (last accessed 28 January 2026)

- National Researchers Assessment: https://www.datos.gov.co/d/bqtm-4y2h (last accessed 28 January 2026)

MinCiencias [51] launched a dashboard titled *La Ciencia en Cifras*: https://minciencias.gov.co/la-ciencia-en-cifras, to explore the raw data contained in the dataset displayed above among other subjects.

### Disciplinary assignment to research groups and researchers

Each researcher is responsible for completing their research portfolio, degrees, experience, and documenting their products on the national platform (CvLAC), specifying their discipline and fields of expertise. The leadership of the respective institution and research group oversees and endorses this entire process, which serves as the primary input for the national assessments of research groups [50]. Consequently, researchers in Colombia are classified into a single discipline of specialization, which reflects their educational background, expertise, and primary field of research [45].

The disciplinary classification of research groups and their affiliated researchers into grand disciplinary areas and disciplines is based on the Organisation for Economic Co-operation and Development (OECD) Fields of Science (FOS) framework [52]. This framework is organized into six major disciplinary areas, which collectively comprise 219 disciplines. The distribution of disciplines across these areas is: Medical and Health Sciences (60), Natural Sciences (48), Engineering and Technology (46), Social Sciences (29), Humanities (22), and Agricultural Sciences (14). This assignment resolves, for example, difficulties in associating authors with a specific discipline that are observed in other co-authorship approaches because authors do not usually declare their discipline in publications [34].

### Sample

To establish the final sample for this study, an initial pool of 7,470 research groups identified in the national calls was filtered according to several criteria. The analysis considers groups that have maintained a stable disciplinary area classification over the evaluation period. Furthermore, the selected groups are required to have a minimum of three members originating from at least two different disciplines. This selection process ensures that changes in the composition of a consistent cohort of groups can be tracked over time. Groups with two members were excluded from the analysis because the calculation of diversity indices for such dyads presents significant methodological limitations. These limitations include statistical instability, which is particularly sensitive to the number of disciplines in small groups; the binary nature of the *Gini coefficient*, which can only reflect perfect equality or maximum inequality; and the absence of a meaningful statistical distribution for such cases. The final sub-sample for this analysis comprised 3,575 research groups.

Table 1 reports the sub-sample of groups by area and rank across 2013–2021. There is a clear upward trend in the total number of research groups, which grew from 1,202 in 2013 to 3,107 in 2021, representing an increase of approximately 158%. All six areas experienced growth, though the rate and scale of this expansion varied. Social sciences exhibited the most dramatic growth, more than tripling from 293 groups in 2013–923 in 2021. This field also became the largest in terms of the absolute number of groups by the end of the period, with an average of ~588 assessed groups per year. Engineering and Technology also showed substantial growth, increasing from 229 to 607 groups. It was consistently one of the largest fields, with an average of ~409 assessed groups per year. Medical and Health Sciences and Natural Sciences both more than doubled their numbers, growing from 196 to 577 and from 300 to 627, respectively, and an average of ~392 and ~456 assessed groups per year, respectively. Humanities and Agricultural Sciences saw steady but more modest increases. The humanities grew from 94 to 220 groups, while agricultural sciences increased from 90 to 153.

Concerning the rank by area, the Social Sciences maintain the highest average number of research groups in nearly every primary rank (A, B, C), while Natural Sciences and Engineering are particularly strong in the higher-level ranks (A1 and A). The Humanities and Agricultural Sciences consistently show the lowest average number of groups across all categories. The highest concentration of groups is found in Rank B and Rank A, led overwhelmingly by the Social

**Table 1. Count of groups by major disciplinary area and national rank.**

| | 2013 | 2014 | 2015 | 2017 | 2019 | 2021 |
|---|---|---|---|---|---|---|
| *Agricultural sciences* | *90* | *107* | *121* | *134* | *145* | *153* |
| A1 | 17 | 16 | 26 | 27 | 35 | 34 |
| A | 10 | 20 | 23 | 24 | 29 | 41 |
| B | 31 | 38 | 30 | 32 | 41 | 35 |
| C | 18 | 31 | 39 | 39 | 32 | 39 |
| Reconocido | 14 | 2 | 3 | 12 | 8 | 4 |
| *Engineering and technology* | *229* | *289* | *354* | *449* | *528* | *607* |
| A1 | 75 | 65 | 87 | 112 | 137 | 168 |
| A | 31 | 38 | 77 | 108 | 145 | 159 |
| B | 63 | 96 | 97 | 110 | 113 | 130 |
| C | 48 | 86 | 89 | 105 | 124 | 135 |
| Reconocido | 12 | 4 | 4 | 14 | 9 | 15 |
| *Humanities* | *94* | *65* | *105* | *159* | *194* | *220* |
| A1 | 9 | 4 | 11 | 10 | 30 | 42 |
| A | 25 | 13 | 23 | 39 | 58 | 60 |
| B | 23 | 20 | 32 | 56 | 54 | 60 |
| C | 25 | 25 | 37 | 47 | 49 | 50 |
| Reconocido | 12 | 3 | 2 | 7 | 3 | 8 |
| *Medical and health sciences* | *196* | *301* | *355* | *434* | *489* | *577* |
| A1 | 54 | 44 | 62 | 84 | 118 | 129 |
| A | 12 | 37 | 56 | 67 | 95 | 114 |
| B | 76 | 116 | 104 | 142 | 128 | 156 |
| C | 41 | 95 | 122 | 129 | 131 | 164 |
| Reconocido | 13 | 9 | 11 | 12 | 17 | 14 |
| *Natural sciences* | *300* | *352* | *426* | *481* | *552* | *627* |
| A1 | 82 | 83 | 94 | 113 | 149 | 152 |
| A | 40 | 72 | 89 | 104 | 138 | 169 |
| B | 86 | 106 | 115 | 123 | 138 | 149 |
| C | 69 | 83 | 109 | 119 | 106 | 129 |
| Reconocido | 23 | 8 | 19 | 22 | 21 | 28 |
| *Social sciences* | *293* | *324* | *456* | *671* | *859* | *923* |
| A1 | 30 | 30 | 58 | 93 | 139 | 189 |
| A | 50 | 62 | 115 | 182 | 244 | 285 |
| B | 96 | 105 | 138 | 190 | 241 | 233 |
| C | 92 | 113 | 129 | 170 | 210 | 194 |
| Reconocido | 25 | 14 | 16 | 36 | 25 | 22 |
| Total (all disciplines) | 1,202 | 1,438 | 1,817 | 2,328 | 2,767 | 3,107 |

Sciences, which average approximately 167 and 156 groups in these ranks, respectively. Rank C is also significant, again dominated by Social Sciences (151 groups) and Medical and Health Sciences (114 groups). Rank A1 shows a different pattern, with Natural Sciences (112 groups) and Engineering and Technology (108 groups) having the highest averages. The *Reconocido* (i.e., recognized or acknowledged) category represents the smallest cohort across all fields, with the Social Sciences having the highest average at 23 groups. For visual aid and synthesis, we added Fig 1 showing

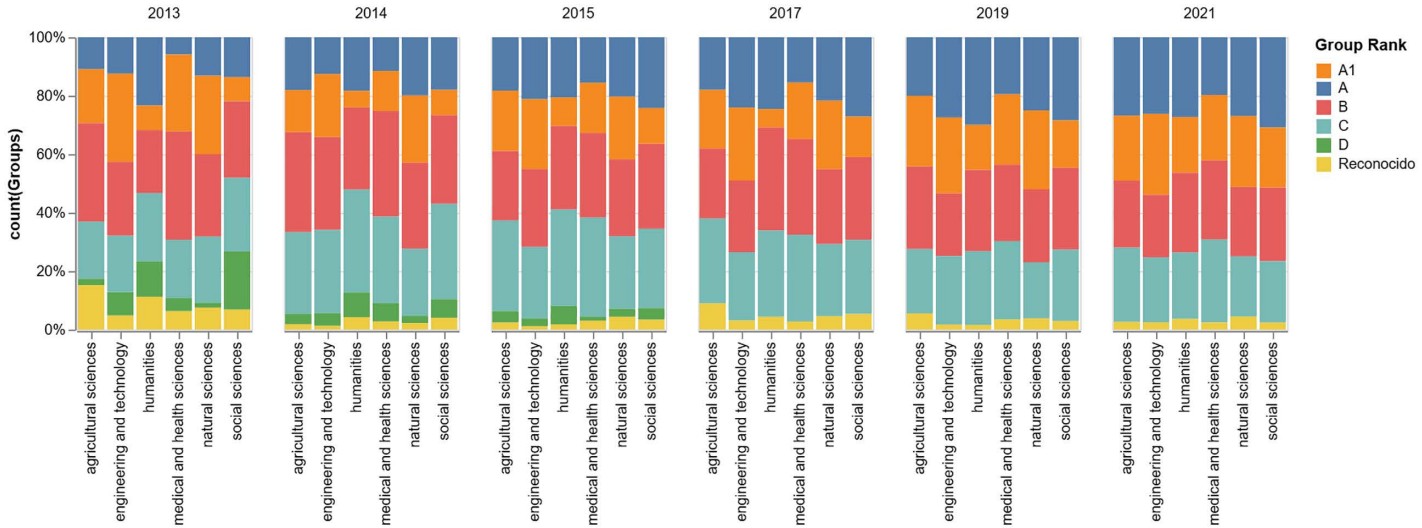

**Fig 1. Percentage of groups by major disciplinary area and national rank per year.**

the percentage of groups by major disciplinary area and national rank per year. Also, in numeral 3.3.2, we will explain the ranks and their relationship with scientific prestige and performance.

We also categorized each group as small (2–5 members), medium (6–10 members), or large (11 or more members) based on team size's importance in producing groundbreaking scientific work [20,53]. Fig 2 displays the Research Group size distribution of all periods. The distribution remains consistent, with an average composition across all periods of approximately 51% small groups, 35% medium groups, and 13% large groups.

To enable a proper analysis, a multi-stage process was employed to assign a singular, uniform disciplinary classification to each of the 29,517 researchers in the original sample. This procedure resolved instances of missing or multiple classifications recorded for individual researchers. The primary criterion for resolving ambiguity is recency, where the discipline associated with the researcher's most recent national call is considered definitive. This approach is based on the assumption that a researcher's focus narrows into a specific cognitive niche as their career matures. If a researcher has missing data, the classification is imputed using their modal (most frequently recorded) discipline. In instances where no single mode can be determined, the process defaults back to the recency criterion, assigning the last valid, non-missing discipline on record. A total of 21,994 researchers were affiliated with the research group sub-sample.

The average percentage of researchers assigned to each major area across all years was: Social Sciences (30%), Natural Sciences (24%), Engineering and Technology (20%), Medical and Health Sciences (16%), Humanities (6%), and Agricultural Sciences (5%). Table 2 shows the median group size by area and rank across 2013–2021. In broader terms, higher-ranked groups, particularly those in Rank A1, have significantly increased in size over time, while lower-ranked groups have maintained a consistently smaller and more stable composition. Rank A1 groups show the most significant and consistent growth across all disciplines. For instance, in Engineering and Technology, the median size of A1 groups grew from 7 to 11 members, and in Medical and Health Sciences, they also expanded from 6 to 11 members. Rank A groups demonstrated more modest growth, generally remaining stable or increasing slightly. In Natural Sciences, for example, they remained consistently between 4 and 6 members. Rank C and *Reconocido* groups remained the smallest and most stable in size over the entire period, typically averaging between 4 and 5 members with minimal fluctuation.

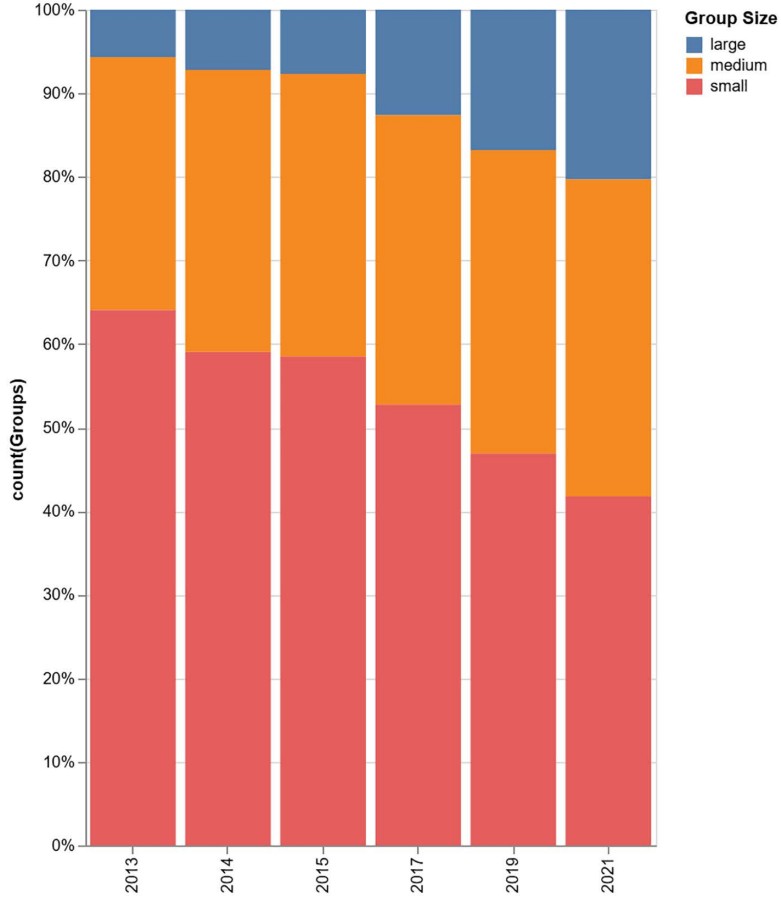

**Fig 2. Group size distribution.**

### Variables

**Operational differences between science teams and research groups in Colombia.** In the field of Science of Team Science (SciTS), a science team is defined as 'two or more individuals with different roles and responsibilities, who interact socially and interdependently within an organizational system to perform tasks and accomplish common goals' [20]. Operationally, in bibliometric studies, teams' size is calculated based on co-authorships. In contrast, a research group in Colombia, our unit of analysis, is defined as follows [50]:

*'The basic modern unit for generating scientific knowledge and its application to technological development, comprising individuals from one or more disciplines and institutions who synergistically work together around a specific field of knowledge'.* (translation by the author).

Unlike the often-transient nature of a co-authorship in research papers, a Colombian research group is a formal and stable entity within a given academic unit (e.g., school or department). These groups consist of a defined number of affiliated researchers, research direction, and designated leadership. A crucial distinction lies in the accreditation of intellectual output; knowledge produced collectively is formally attributed to the research group, regardless of whether it was created by group members alone or in collaboration with external co-authors. Consequently, an external collaborator on a publication

**Table 2. Median groups size (number of members) by major disciplinary area and national rank.**

| | 2013 | 2014 | 2015 | 2017 | 2019 | 2021 |
|---|---|---|---|---|---|---|
| *Agricultural sciences* | *5.2* | *5.4* | *5.7* | *6.3* | *6.2* | *6.1* |
| A1 | 8 | 8 | 9 | 11 | 11 | 11 |
| A | 5 | 6.5 | 6 | 6.5 | 6 | 7 |
| B | 5 | 5 | 5.5 | 5 | 6 | 5 |
| C | 4 | 4 | 4 | 4 | 4 | 4 |
| Reconocido | 4 | 3.5 | 4 | 5 | 4 | 3.5 |
| *Engineering and technology* | *4.7* | *5.4* | *5.2* | *5.7* | *6* | *6.7* |
| A1 | 7 | 7 | 8 | 8 | 9 | 11 |
| A | 4 | 6 | 6 | 6 | 7 | 7 |
| B | 5 | 5 | 5 | 5.5 | 6 | 6.5 |
| C | 4 | 4 | 4 | 5 | 4 | 5 |
| Reconocido | 3.5 | 5 | 3 | 4 | 4 | 4 |
| *Humanities* | *4.6* | *4.6* | *5* | *5.4* | *5.4* | *5.6* |
| A1 | 5 | 6.5 | 6 | 9 | 7 | 9 |
| A | 5 | 5 | 6 | 5 | 6 | 6 |
| B | 5 | 4.5 | 5 | 4 | 6 | 5 |
| C | 4 | 4 | 3 | 4 | 4 | 4 |
| Reconocido | 4 | 3 | 5 | 5 | 4 | 4 |
| *Medical and health sciences* | *4.3* | *5.6* | *5.8* | *5.8* | *6* | *6.6* |
| A1 | 6 | 8 | 9 | 9 | 10 | 11 |
| A | 3.5 | 6 | 5.5 | 6 | 6 | 7 |
| B | 4 | 5 | 5.5 | 5 | 6 | 6 |
| C | 4 | 4 | 4 | 4 | 4 | 4 |
| Reconocido | 4 | 5 | 5 | 5 | 4 | 5 |
| *Natural sciences* | *4.4* | *5.2* | *5* | *5.3* | *5* | *6.3* |
| A1 | 6 | 7 | 7 | 8 | 9 | 11 |
| A | 4 | 5 | 5 | 5 | 5 | 6 |
| B | 4 | 5 | 5 | 5 | 4 | 5 |
| C | 4 | 4 | 4 | 4 | 3 | 4 |
| Reconocido | 4 | 5 | 4 | 4.5 | 4 | 5.5 |
| *Social sciences* | *5.2* | *5.5* | *5.2* | *6.1* | *6* | *6.5* |
| A1 | 7 | 7.5 | 9 | 11 | 10 | 11 |
| A | 6 | 6 | 5 | 7 | 7 | 8 |
| B | 5 | 5 | 5 | 5 | 5 | 5 |
| C | 4 | 4 | 4 | 4 | 4 | 4.5 |
| Reconocido | 4 | 5 | 3 | 3.5 | 4 | 4 |

does not become a formal member of the group, which fundamentally differentiates this organizational structure from a team defined solely by co-authored research papers.

**Scientific prestige as scientific performance.** The Colombian classification system for research groups and researchers employs a ranking structure. Research groups are classified into five ranks: A1, A, B, C, and *Reconocido*; A1 representing the highest rank. These rankings are determined by research outputs, collaboration networks, and contributions to the scientific community. The requirements for the classification of research groups are specified on pages 128–131 of the official MinCiencias research group model. This document details the complete list of criteria necessary for

a group to be assigned to a specific category: MinCiencias [48]. In essence, the rank is a hierarchical classification where the A1 rank represents an elite rank distinguished by a rigorous double filter, requiring concurrent placement in the top quartile (Q1) for both Group and Top Product indicators. In contrast, ranks A and B occupy broader competitive ranges— encompassing the top 50% and 75% respectively—and mandate only the presence of high-quality products rather than specific quartile positioning for those outputs. Finally, C rank operates distinctively as a non-competitive baseline based on a checklist of minimum requirements, such as existence duration and human resource formation; this design positions it as a transitional entry point intended to foster advancement toward consolidation, thereby negating the need for internal subdivisions like C1 or C2, as well as in the B rank.

The A1 rank is distinguished by research outputs of high impact and strong international collaboration. These groups also demonstrate significant leadership in training new researchers and exhibit a high level of engagement in technological and social innovation. Following the top rank, groups in A rank produce consistent research with a notable national impact. They are active participants in academic networks and maintain a strong role in the training of researchers. The subsequent rank, B, is for groups on a growing trajectory in their research outputs, showing moderate levels of collaboration and innovation activities. Finally, C rank encompasses emerging groups with foundational research outputs. Their focus is primarily on local impact as they are in the early stages of developing their research capabilities. An additional rank, *Reconocido*, is designated for groups that participate in the annual call and are acknowledged by MinCiencias but do not meet the criteria for the other classifications [50]. D rank was discontinued after the first years analyzed and is therefore excluded from this analysis. The rank of the research group serves as the variable for scientific status, which reflects research group performance. Moreover, this strategy, grounded in research group evaluations of various factors, expands the range of variables used to determine scientific prestige, moving past simple citation analysis, a prevalent bibliometric tool that might not fully account for the dynamic nature of scientific influence [34].

**Trajectory.** To analyze the longitudinal performance in the national ranking of research groups, a categorical trajectory variable was developed based on the formal ordinal hierarchy of ranks. First, a group's status was determined by comparing its rank to that of the previous period. This comparison resulted in four distinct classifications: *Advancement*, if there was an improvement in rank; *Decline*, if the group dropped in rank; *Stagnation, if there was* no change in rank; or *New*, for the groups that first recorded appearance in the dataset. This procedure produces a discrete variable that captures the annual directional change for each group.

Subsequently, a longitudinal summary classification was created to provide an aggregate measure of each group's dominant trajectory over the entire observation period. This classification was derived using a modal aggregation, where the most frequently occurring annual status was identified for each research group. Therefore, a group was classified as Advancement if its most frequent status was upward movement in the national ranking; Decline when the group descended in the ranking most of the time; and if no single status was uniquely dominant due to a tie in frequency, the group was classified as Volatile to signify a trajectory of instability. The New status was excluded from this calculation because it represents an initial state rather than a performance change. Table 3 reports an example for each trajectory type.

**3.3.4 Calculating *DIV* for research groups disciplinary diversity.** As we mention in the Introduction section, the *DIV* indicator offers an already tested approach to measure disciplinary diversity [34]. The indicator and its components were therefore adapted to assess the disciplinary diversity among members of research groups, following diversity essential features: *variety, balance, and disparity.* In the context of this framework, *variety* is defined as the number of distinct disciplines represented within a research group from a certain major disciplinary area. This is measured as *Relative variety* (*RV*). For each research group $i$, we calculate the proportion of unique disciplines present relative to the total pool of disciplines identified across all groups. We formalized this as follows (Equation 1):

$$RV_i = \frac{n_i}{N}$$

(1)

**Table 3. Examples of group trajectory types.**

| Trajectory type | Group Code | Year | Rank | Year | Status |
|---|---|---|---|---|---|
| Advancement | COL0000238 | 2015 | B | 2015 | New |
| | | 2017 | A | 2017 | Advancement |
| | | 2019 | A | 2019 | Stagnation |
| | | 2021 | A1 | 2021 | Advancement |
| Decline | COL0000659 | 2013 | Reconocido | 2013 | New |
| | | 2014 | A1 | 2014 | Advancement |
| | | 2015 | A | 2015 | Decline |
| | | 2017 | C | 2017 | Decline |
| | | 2019 | A | 2019 | Advancement |
| | | 2021 | Reconocido | 2021 | Decline |
| Stagnation | COL0000049 | 2013 | B | 2013 | New |
| | | 2014 | B | 2014 | Stagnation |
| | | 2015 | B | 2015 | Stagnation |
| | | 2017 | C | 2017 | Decline |
| | | 2019 | C | 2019 | Stagnation |
| | | 2021 | C | 2021 | Stagnation |
| Volatile | COL0001816 | 2013 | D | 2013 | New |
| | | 2015 | C | 2015 | Advancement |
| | | 2017 | C | 2017 | Stagnation |
| | | 2019 | Reconocido | 2019 | Decline |

Where $n_i$ represents the number of distinct disciplines within research groups $i$ and $N$ represents the total number of distinct disciplines appearing across the entire sample.

The second component, *balance*, is defined as the evenness of the distribution of researchers across the disciplines represented within a research group from a certain major disciplinary area. We measured this via *Gini coefficient* ($G$). We formalize this as follows (Equation 2):

$$G = \frac{\sum\limits_{i=1}^{n}\sum\limits_{j=1}^{n}|x_i - x_j|}{2n^2\overline{x}}$$

(2)

Where $x_i$ represents the number of researchers in discipline $i$, and $n$ is the total number of disciplines in a research group from a certain major disciplinary area, and $\overline{x}$ is the arithmetic mean of the number of researchers across all $n$ disciplines. If the $G$ of a group has equal representation across all disciplines it be lower (i.e., high balance), whereas if researchers are concentrated in a few disciplines (i.e., imbalance) the coefficient will be higher.

The third and final component, *disparity*, refers to the conceptual distance between researchers disciplines within a group of a certain major disciplinary area. We computed this via *Average disparity* ($AD$). To achieve this, we follow the standard approach using cosine similarity [26].

A group-by-discipline matrix was first constructed, where the rows represent individual research groups and the columns represent the disciplines. Each cell in this matrix, denoted as $x_{g,d}$, represents the number of researchers in group $g$ affiliated with discipline $d$. From this initial matrix, a square discipline-by-discipline co-occurrence matrix was derived. In this second matrix, both the rows and columns represent the disciplines. Each entry $M_{i,j}$ in the matrix indicates the total number of times disciplines $i$ and $j$ co-occur across all research groups (Equation 3):

$$M_{i,j} = \sum_g (x_{g,i} \times x_{g,j}) \tag{3}$$

Where $x_{g,i}$ and $x_{g,j}$ are the researchers in group $g$ for disciplines $i$ and $j$, respectively.

The cosine similarity between any two disciplines, $i$ and $j$, is calculated using the following equation (Equation 4):

$$\text{Cosine similarity}\,(i,j) = \frac{\sum_g (x_{g,i} \times x_{g,j})}{\sqrt{\sum_g (x_{g,i})^2} \times \sqrt{\sum_g (x_{g,j})^2}} \tag{4}$$

The numerator is the co-occurrence of disciplines $i$ and $j$ across all groups and the denominator normalizes by the product of the magnitudes of the discipline vectors.

The resulting cosine similarity score is then converted into a measure of disparity between disciplines $i$ and $j$ using the following formula (Equation 5):

$$\text{Disparity}(i,j) = 1 - \text{Cosine similarity}(i,j) \tag{5}$$

Finally, for each research group composed of $n$ distinct disciplines, the Aggregated Disparity ($AD$) is calculated using the following equation (Equation 6):

$$AD = \frac{\sum_{i \neq j} \text{Disparity}(i,j)}{n(n-1)} \tag{6}$$

The numerator sums the pair-wise disparities among all distinct discipline pairs within the group, and the denominator represents the total number of unique discipline pairs, ensuring the average is taken correctly.

Finally, we calculate the $DIV$ for each research group as follows (Equation 7):

$$DIV = RV \times (1 - G) \times AD \tag{7}$$

We computed $DIV$ annually, as the number of groups and researchers changes with each national assessment, affecting their corresponding areas, disciplines, and their variety, balance, and disparity. In essence, a group of a certain major disciplinary area with a high $DIV$ score is characterized by a greater number of distinct disciplines, an even distribution of researchers among them, and significant conceptual distance between the disciplines of its members. Conversely, a group with a low $DIV$ score, indicating lower interdisciplinarity, has fewer disciplines, which are more concentrated and conceptually similar to one another. Fig 3 displays both scenarios of groups with higher and lower $DIV$ and Table 4 reports the research groups with the highest and lowest $DIV$ score in 2021.

## 4 Results

Fig 4 illustrates the temporal evolution of the median DIV of research groups across areas and size types for the period between 2013 and 2021. Also, the S1 Table reports in more detail the median DIV by area, group rank, and size. For the sake of interpretability, we excluded the *Reconocido* rank (<5% of the total sample).

Diversity trends varied significantly across research areas between 2013 and 2021. In most fields—specifically Medical and Health Sciences, Engineering and Technology, Natural Sciences, and Humanities—the median diversity index (DIV) in 2021 decreased compared to 2013 levels. Conversely, Agricultural Sciences and Social Sciences exhibited a marginal increase in median DIV over the same period. When analyzing trends by group classification, higher-ranked groups (A1

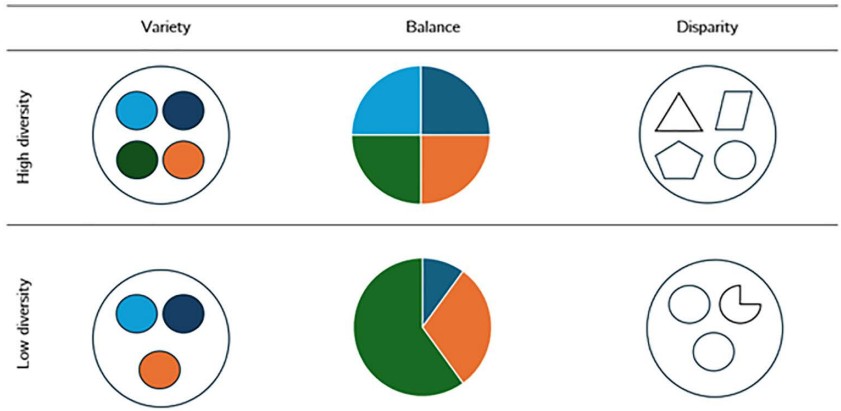

**Fig 3. Cases of research groups with higher and lower diversity.** Elaboration of the author based on Ref. [14]. Note: Variety is indicated by the number of distinct elements, each distinguished by specific color coding. Balance is represented by the proportional color composition; the dominance of a single hue denotes an uneven distribution or concentration of researchers within that discipline. Disparity is visualized through geometric shapes corresponding to major disciplinary areas—such as circles for Natural Sciences and triangles for Social Sciences—which illustrate the cognitive distance between the disciplines.

and A), which are characterized by a larger median number of researchers, have recently shown an upward trajectory in diversity, whereas groups in ranks B and C have generally declined.

Despite these temporal trends, the relationship between a research group's disciplinary diversity and its national rank is non-linear and moderated by factors such as scientific area and group size. While higher-ranked groups (A1 and A) display increasing trajectories, the highest overall median disciplinary diversity is observed in B-ranked groups, followed by ranks A and C. Consequently, the aggregated data does not support a simple direct correlation between greater disciplinary diversity and higher scientific prestige. Instead, field-specific patterns emerge: a positive association between diversity and prestige is prominent in the Humanities and Medical and Health Sciences. In contrast, an inverse relationship characterizes the Social Sciences, Natural Sciences, Agricultural Sciences, and Engineering and Technology, where groups exhibiting higher diversity are more frequently found in lower national ranks.

Fig 5 displays the percentage of groups by area, rank and size. Larger groups are overly represented in the highest ranks of academic performance. This trend is most pronounced in the fields of Engineering and Technology, Medical and Health Sciences, and Social Sciences. Within these disciplines, although large research groups constitute a smaller proportion of the total, they are overrepresented in the A1 rank.

In Engineering and Technology, large groups constitute a minor fraction of the total number of research units, representing ~3% in 2021. However, they comprise a significant percentage of the groups in the A1 rank, at ~13% for the same year. Conversely, small groups are the most numerous in this field, with an average of ~9% per year, yet they are concentrated in the lower performance ranks. A similar pattern is observed in the Natural, Agricultural, and Medical and Health Sciences. This trend is particularly notable in the Natural Sciences, which has the largest number of small groups overall, averaging ~13% annually.

The Social Sciences and Humanities are characterized by a predominance of small and medium-sized groups. Within Social Sciences, the largest disciplinary area by volume, small groups are particularly numerous, averaging ~14% per year. A detailed examination reveals, however, that the few large groups in this field are highly successful, accounting for ~13% of the A1 rank in 2021. In the Humanities, the number of large groups is almost negligible. Consequently, this disciplinary area is dominated by small groups, which are distributed across all ranks, including the highest ranks.

**Table 4. Example of a research group with the highest and lowest DIV, 2021.**

| Type | Group code | DIV | Year | Disciplinary area | Rank | Researcher ID | Researchers' discipline |
|---|---|---|---|---|---|---|---|
| Highest *DIV* | COL0099642 | 0.0363 | 2021 | Medical and health sciences | A1 | 11925 | Biochemistry and molecular biology |
| | | | | | | 43575 | Other clinical medicine issues |
| | | | | | | 48084 | Infectious diseases |
| | | | | | | 202997 | Immunology |
| | | | | | | 205419 | General and internal medicine |
| | | | | | | 205818 | Rheumatology |
| | | | | | | 252794 | Other biology |
| | | | | | | 263664 | Epidemiology |
| | | | | | | 407480 | Health care sciences and services |
| | | | | | | 727806 | Orthopedic |
| | | | | | | 767283 | Nutrition and diets |
| | | | | | | 816353 | Genetics and inheritance |
| | | | | | | 955191 | Radiology, nuclear medicine and imaging |
| | | | | | | 1325132 | Psychology |
| | | | | | | 1362886 | Other clinical medicine issues |
| | | | | | | 1364136 | Other clinical medicine issues |
| | | | | | | 1369397 | Transplants |
| | | | | | | 1428717 | Epidemiology |
| | | | | | | 1438255 | Anatomy and morphology |
| | | | | | | 1473805 | Human genetics |
| | | | | | | 1475978 | Biomedical socio-sciences |
| | | | | | | 1476266 | Genetics and inheritance |
| | | | | | | 1492639 | Epidemiology |
| | | | | | | 1512936 | Public health |
| | | | | | | 1567207 | Health-related biotechnology |
| Lowest *DIV* | COL0044878 | 0.0010 | 2021 | Social sciences | A1 | 35701 | Psychology (includes man-machine relationships) |
| | | | | | | 525480; 568325; 568341; 568406; 568414; 723851 | Psychology (includes learning therapies, speech, visual and other physical and mental disabilities) |

Fig 6 displays the percentage of research groups by trajectory status. After excluding groups that appeared only once, the data show that a majority exhibited a stagnation trajectory (~54%). This was followed by groups classified as volatile (~22%) and those showing advancement (~18%). The smallest category consisted of groups that declined in rank (~5%).

It is important to highlight that being classified as a group in stagnation is not necessarily a bad verdict. In Fig 7, we reported the composition by rank according to the group's trajectory across periods. An analysis of the four trajectory categories from 2013 to 2021 reveals distinct patterns of movement within the classification system.

Groups in the Advancement category, which were predominantly in the lower ranks, demonstrated a clear upward progression. By 2021, a majority of these groups had achieved the A and A1 ranks. This confirms that groups whose most frequent action was to improve their classification did successfully ascend to the top of the system over the observed period. Conversely, groups in the Decline category showed the opposite trend. While these groups had a significant presence in the higher ranks (A, A1) in 2013, by 2021, the lower classifications of C and *Reconocido* became dominant, and the proportion of groups in the top ranks diminished. The Stagnation category reveals a key insight: this classification

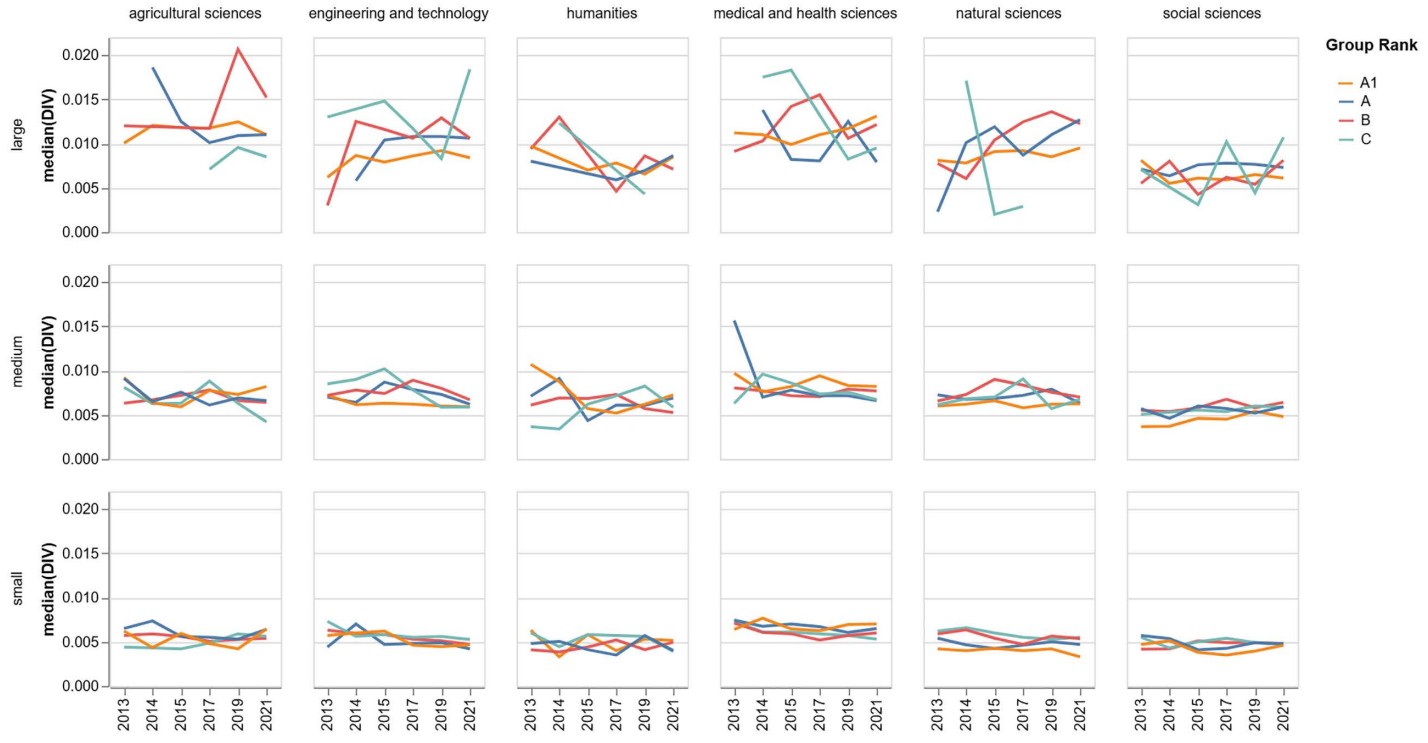

**Fig 4. Median DIV of research groups across areas and size types for the period between 2013 and 2021.**

does not necessarily equate to being *stuck* at a low rank. Although these groups were spread across all classifications in 2013, the proportion in the top ranks (A, A1, and B) grew over time as the share in the lowest ranks decreased. This suggests that many groups in this category are those that successfully maintained a high rank year after year, rather than those unable to improve from a low one. Finally, the Volatile category also exhibited a general upward trend. The lowest classifications, prominent in 2013, decreased significantly by 2021. This indicates that even groups with inconsistent year-to-year performance experienced a net improvement in their classification over the long term.

Concerning the relationship between disciplinary diversity, research group trajectory, and size, Fig 8 reports that Engineering and Technology and Agricultural Sciences are the only disciplinary areas with no groups with *decline* trajectory. No clear relationship is apparent between a group's *advancement* trajectory and its median *DIV*. In most cases, the median disciplinary diversity for groups in the *advancement* trajectory is similar to that of groups in the *stagnation* trajectory. An exception is noted in the Social Sciences, where the median diversity of the *stagnation* group is higher than that of the *advancement* group. Despite this variation, a recurring pattern across all disciplinary areas is that groups in the *decline* trajectory consistently present the lowest median diversity among all trajectory categories.

For a more reliable and detailed analysis, we used a Kruskal-Wallis H-test to compare the median *DIV* across the four trajectory groups: Advancement, Decline, Stagnation, and Volatile. The S1 Table, reports in more detail the median DIV by area, group rank, and size. The results showed a statistically significant difference in median diversity levels among the trajectory groups, $\chi^2(3, N = 12,913) = 104.09$, $p < .001$. To isolate the specific sources of this difference, Dunn's post-hoc test with a Bonferroni correction was performed for pairwise comparisons (Table 5). The analysis revealed significantly different median *DIV* in all groups ($p < .001$), except between the Advancement and Volatile groups ($p = 1.00$), suggesting their median *DIV* is statistically similar. However, the magnitude of the difference between trajectory groups

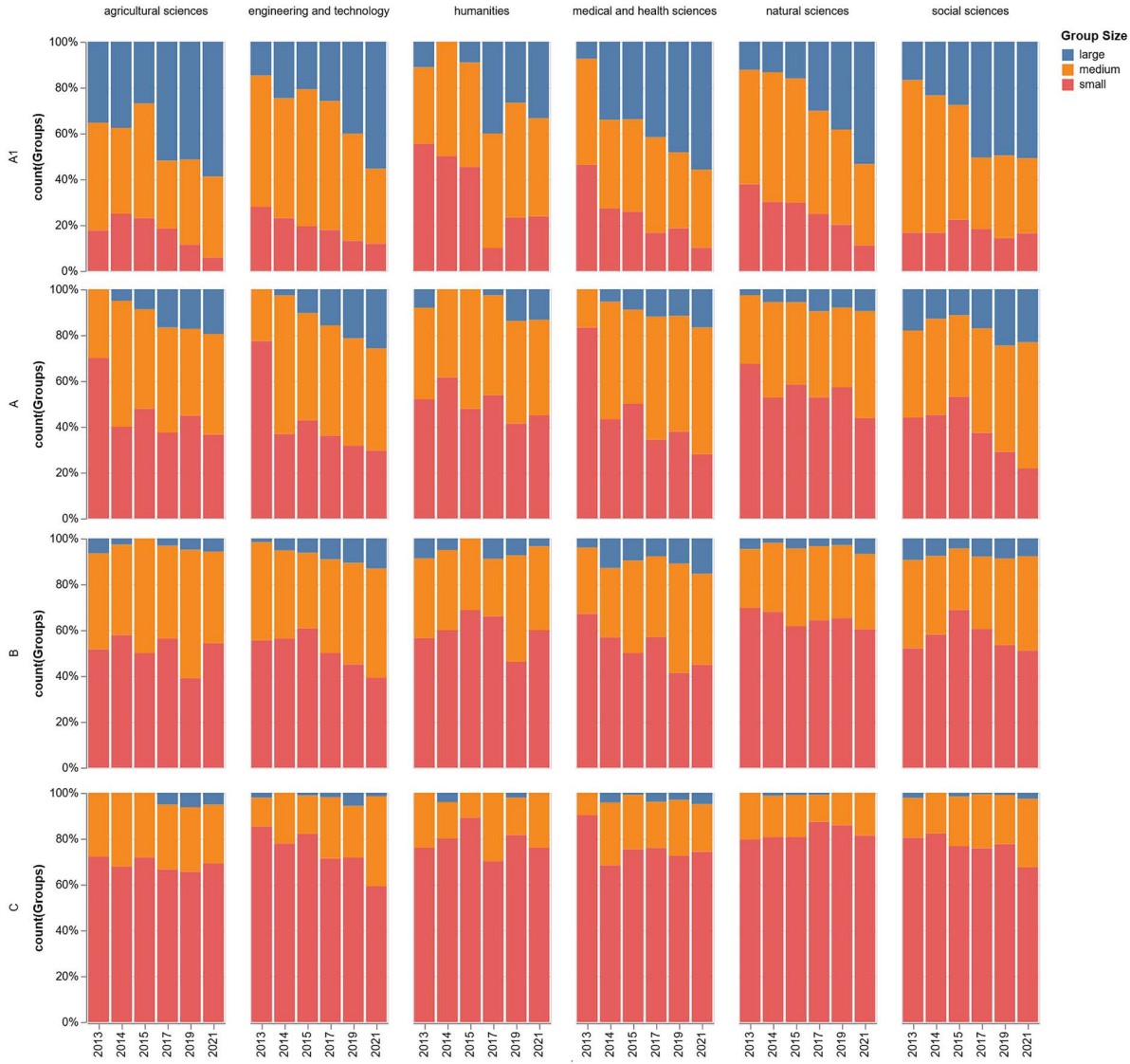

**Fig 5. Percentage of groups by area, rank and size.**

and median *DIV* was small, with an epsilon-squared effect size of .008. In essence, a group's trajectory explains less than 1% of the variance in its *DIV* score. Besides this, an interesting takeaway is that research groups on an Advancement path have a diversity score that is statistically identical to groups with a Volatile trajectory, suggesting that the process of achieving major scientific prestige is intrinsically linked to instability. In contrast with the other trajectory groups, a state of Stagnation or Decline is associated with a more rigid or specific diversity structure, unlike the disciplinary diversity seen in Advancement and Volatile trajectories.

## 5 Discussion and conclusion

This study aimed to quantify through six periods the disciplinary diversity trajectory across all research groups in Colombia and analyze its relationship with scientific prestige. We deepened our understanding of the disciplinary diversity within

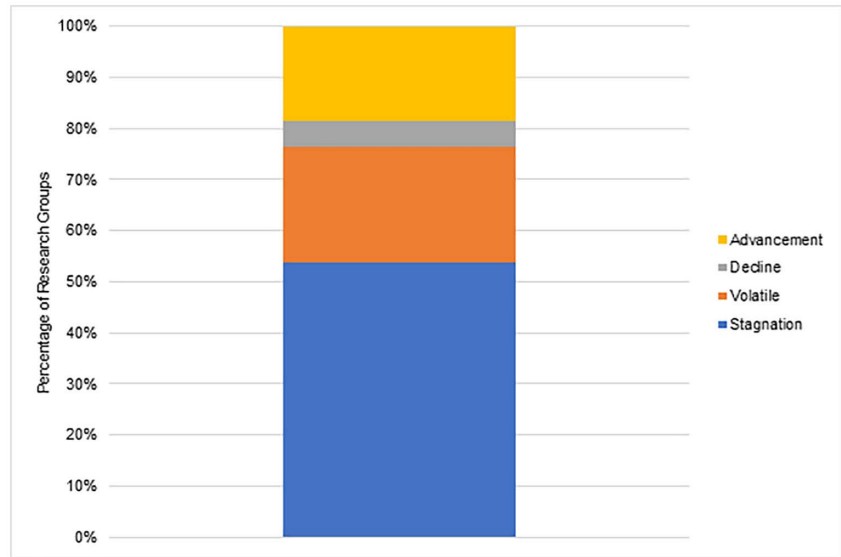

**Fig 6. Percentage of groups by trajectory status.**

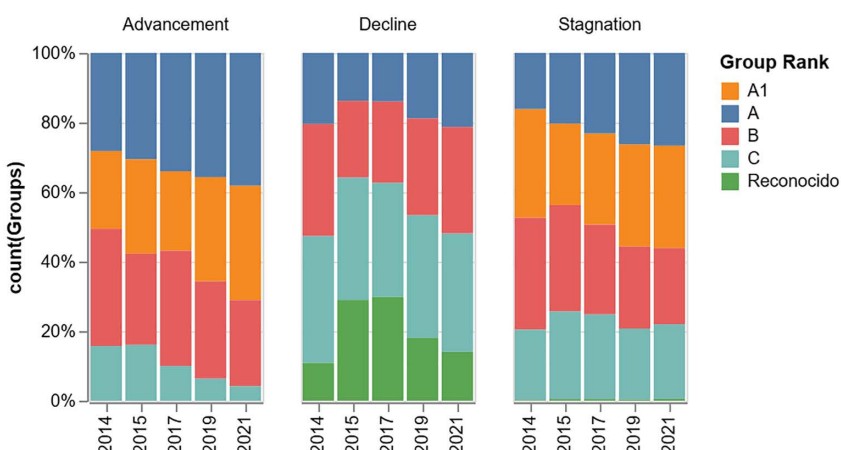

**Fig 7. Composition of research group trajectory type by rank.**

research groups and their performance over time, specifically in terms of scientific reputation reflected by the ranking results of the national assessment. This analysis used official public data from government organizations like MinCiencias, in the context of Colombia, the Global South. In broad terms, the disciplinary diversity of research groups is associated with their primary disciplinary area, size, and academic rank. The relationship between disciplinary diversity and scientific prestige and trajectory is non-monotonic. Although this relationship implies that outcomes vary significantly at different levels of diversity, the size of the effect is not substantial.

We found that inMedical and Health Sciences and Natural Sciences, as well as in Engineering and Technology and Humanities, the median *DIV* in 2021 was lower than in 2013. Meanwhile, Agricultural and Social Sciences showed a marginal increase over the same period. Quantitative assessments of disciplinary diversity (i.e., IDR), across different ways to organize knowledge (e.g., sciences, disciplines, area, fields, topics, etc.) vary considerably depending on the

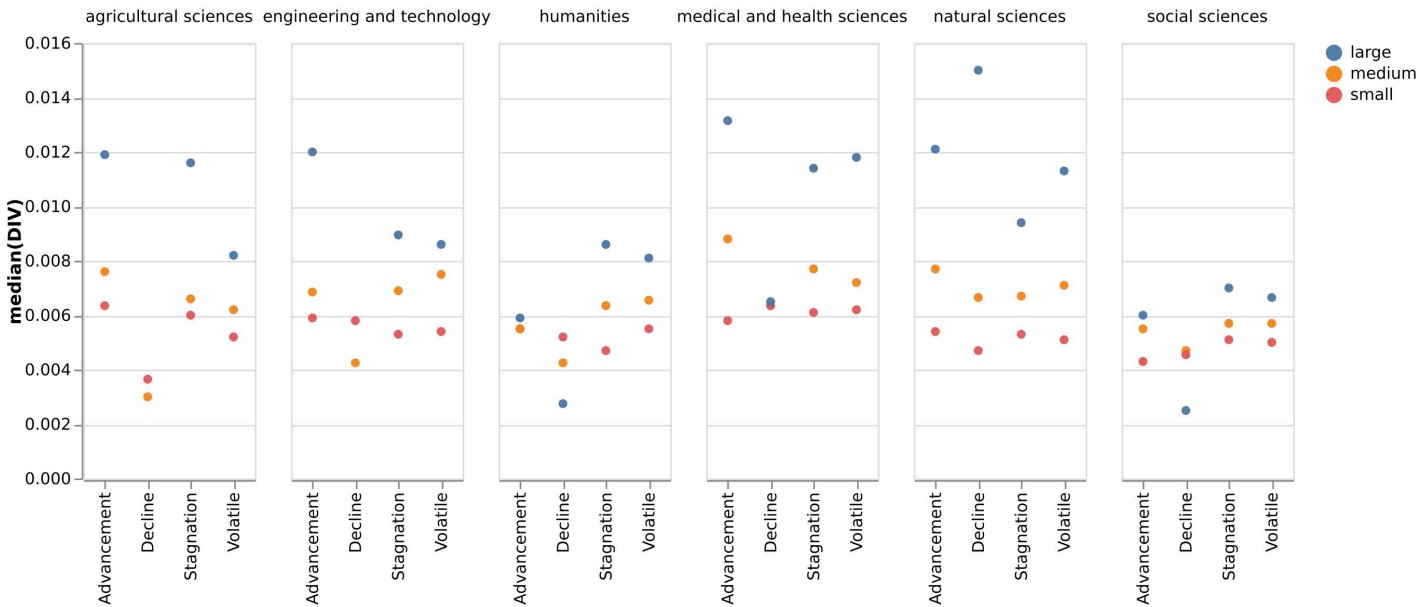

**Fig 8. Median DIV, research group trajectory, and size.**

**Table 5. Pairwise Comparisons of Median Diversity (DIV) 2013-2021 across research group trajectories using Dunn's Test with Bonferroni Correction.**

| Comparison between group trajectories | $n_1$ | $n_2$ | $Z$ | Adjusted $p$-value |
|---|---|---|---|---|
| Advancement vs. Decline | 1,909 | 502 | −5.51 | <.001 |
| Advancement vs. Stagnation | 1,909 | 7,084 | 4.91 | <.001 |
| Advancement vs. Volatile | 1,909 | 2,787 | 0.12 | 1.000 |
| Decline vs. Stagnation | 502 | 7,084 | 8.73 | <.001 |
| Decline vs. Volatile | 502 | 2,787 | 5.78 | <.001 |
| Stagnation vs. Volatile | 7,084 | 2,787 | −5.51 | <.001 |

theoretical and methodological approach used for the analysis [54]. For example, methods that rely on citation analysis of reference lists indicate that fields such as biology and chemistry exhibit high levels of diversity [55], whereas analyses of co-authorship data suggest that Medicine features the most frequent and recurring collaborations between disciplines [56], and also the perceived interdisciplinarity of a single field like Environmental Sciences can be inconsistent across different metrics, appearing interdisciplinary based on project grant text while seeming mono-disciplinary when assessed using article references or abstracts [57].

The Social and Agricultural Sciences featured the highest prevalence of research groups comprising members from disciplines with significant cognitive distance and an internally unbalanced composition when compared to the other major areas studied. It is also relevant to consider the degree of specialization in other disciplinary areas, excluding the Humanities, which are subdivided into a greater number of fields; for example, Medical and Health Sciences contain 60 disciplines, while Natural Sciences and Engineering and Technology have 48 and 46, respectively. This high degree of specialization may be related to the increasing focus on the frontiers of knowledge areas such as in Medicine, Physics, and Chemistry [58]. Furthermore, research policy in Colombia does not contain explicit guidelines or incentives for treating

the disciplinary diversity of research groups as a positive criterion in their national assessment and ranking. The 2021 research group call (275 pages) contains no explicit reference to diversity or interdisciplinarity in the evaluation criteria, either in the productivity nor their group's composition [32,59].

When refining by rank, we showed that higher-ranked groups (A1 and A) are larger and exhibit greater disciplinary diversity. Over time, diversity in top-ranked groups has trended upward, whereas it has generally declined for mid-ranked groups. The expectation of heightened diversity in larger groups, consistent with earlier findings [56], stems from the greater potential for including researchers from a range of disciplines as group size expands. However, the relationship between a group's disciplinary diversity and its national rank is non-linear and is moderated by the scientific area and group size. Our analysis showed that groups ranked B have the highest $DIV$. This relationship is major area-dependent; since it is positive in the Humanities and Medical and Health Sciences but appears to be inverse in fields like Social Sciences and Engineering and Technology, where more diverse groups are often in lower ranks. Unlike scientific impact measured via citations, diversity is context-specific thereby providing a second dimension to the evaluation, as stated by Zhang et al. [57] when applying $DIV$ to universities research portfolios.

Analysis of group composition shows that larger research units are overrepresented in the highest ranks of academic performance, a trend most pronounced in Engineering and Technology, as well as in Medical and Health Sciences and Social Sciences. Although large groups are a minority, they comprise a significant percentage of the top-ranked A1 rank in these fields. In contrast, small groups are more numerous overall but are concentrated in lower performance ranks, with the Humanities being an exception where small groups are distributed across all ranks. Colombia's science system, although one of the most productive in Latin America and the Caribbean and having achieved several notable break-throughs, does not generate a significant figure of globally influential science [60,61]. This discrepancy stems from both historical and institutional factors: a traditional focus on graduating professionals rather than training researchers and scientists, coupled with a lack of governmental prioritization for science and insufficient job-market incentives to encour-age scientific careers [62–66]. Consequently, the research landscape is shaped by these constraints, leading top-tier groups to reinforce less-transformative paradigms, consolidating more traditional scientific subjects [67]. This conservative approach aligns with the standards of MinCiencias for knowledge generation and influence. Furthermore, the national science system lacks financial and structural resilience to support high-risk scientific programs, compelling a focus on more predictable and incremental research agendas rather than transformative, risk-taking endeavors [23]. Additionally, a crucial aspect of larger groups is their ability to generate a greater volume of knowledge, more likely to be recognized in prestigious journals and future citations, stemming from the increased number of researchers within the group, all pooling social and financial capital, as well as know-how [68–70].

While group size and disciplinary diversity are concurrent attributes of successful research teams, their impact on scientific prestige is mediated by different processes. The positive correlation between group size and ranking is predom-inantly influenced by the attainment of critical mass, wherein larger entities leverage the consolidation of social and finan-cial resources, alongside expertise, to foster increased knowledge output and collaboration that is quantifiable by national evaluation standards [70–72]. In this context, the diversity of disciplines arises in part as a structural consequence of expansion, as larger groups encompass a broader spectrum of fields, rather than being the singular independent deter-minant of prestige. However, size significantly moderates the effect of diversity. This indicates that while large groups possess the structural resilience to harness the benefits of cognitive heterogeneity, smaller groups may face *coordination penalties* where the costs of integrating disparate disciplines outweigh the potential benefits, particularly in the absence of the resource buffers found in larger groups.

Regarding group trajectories, most groups exhibited Stagnation, followed by Volatility and Advancement. The Stagna-tion category includes many high-performing groups that successfully maintained their top rank over time. These findings are another support for the *Matthew effect* in science, where prestige in the scientific community is a self-reinforcing phenomenon [73,74]. This creates a cycle in which established and reputable scientists and teams continue to accrue

advantages that solidify their standing, both at the researcher-level as well as the institutional level [75]. For instance, in the analysis of individual researchers in Colombia, the strongest predictor of career progression within the national ranking system was the researcher's previous rank [76], a similar pattern is observed here.

Although no clear relationship was found between a group's advancement trajectory and its median disciplinary diversity, groups categorized in a Decline trajectory systematically presented the lowest median diversity, meaning that while high diversity is not a clear prerequisite for advancement, low diversity is a common feature of declining research groups. The Kruskal-Wallis H-test reported a statistically significant difference in median $DIV$ among the trajectory groups ($p < .001$), though the effect size was small. Post-hoc analysis revealed that the median $DIV$ for groups in the Advancement trajectory is statistically similar to that of groups with a Volatile trajectory. This suggests the process of achieving scientific prestige is associated with a diversity structure similar to that of unstable groups. Despite disciplinary diversity being viewed as a valuable opportunity that expanded access to stakeholder networks and fostered a cross-learning environment, it also introduced challenges in aligning objectives, synchronizing tasks, and integrating disparate methodological approaches, institutional priorities, and further disciplinary traditions (e.g., disparity among the relevance of specialized journals among members) [56,77]. In addition, as shown by Yang et al. [42], higher disciplinary diversity is associated with lower levels of scientific disruption since this diversity seems to be invested in consolidating existing knowledge rather than producing disruptive innovations. These findings can vary in analyses of disciplinary diversity within a specific field, such as Artificial Intelligence. In that context, the diversity of research interests was strongly associated with the innovative performance of interdisciplinary teams. This suggests that the ability to blend and reorganize knowledge, vital for innovation, leads to better team outcomes within one field, as long as it stays balanced to avoid a decline in social impact after a certain peak [78].

Harnessing the benefits of disciplinary diversity in science teams and research groups relies on factors here not explored, such as communication, conflict management, and leadership, since the coordination costs associated with bridging disciplinary gaps are significant. Cases have found that successful cross-disciplinary groups overcome these coordination costs through a process of cultural and disciplinary assimilation [14]. Also, funding and institutional support function as key mediators that influence how these teams form and function. For instance, strong organizational models structured as consortia provided the necessary scaffold for deep cross-disciplinary integration but also pressure to secure resources, thereby promoting the *convergence shortcuts*, where teams opt for polymathic generalists rather than facing the coordination challenges of integrating distinct specialists, suggesting that the race to secure funding might incentivize expedient teaming strategies over the more difficult, yet potentially more transformative, work of deep cross-disciplinary coordination [13].

While this study relies on the Colombian context and data, the structural constraints identified—such as the prevalence of incremental research agendas driven by funding insecurity and a historical emphasis on professional training over scientific inquiry—are not unique to Colombia, but also, other Latin American countries [62,79]. Like other emerging economies, the pressure to secure resources in a scarcity-driven environment incentivizes conservative strategies where groups prioritize volume and consolidated topics over high-risk research, which often requires disciplinary diversity [13,14]. Furthermore, the confirmation of the Matthew effect in our trajectory analysis suggests that the stratification of research groups is a systemic feature of metric-based national evaluations, applicable to other countries employing similar ranking mechanisms. Finally, the non-monotonic relationship between disciplinary diversity and scientific prestige underscores a universal SciTS principle that requires further resources, strategies, and approaches to understand it in depth [44].

Regarding the policy implications, Colombian policymakers should consider incorporating metrics of disciplinary diversity into their evaluation criteria, given that low diversity is consistently associated with declining research groups in the national ranking system. However, these findings must be interpreted with caution. The relationship between diversity and prestige is non-monotonic with a small effect size, indicating that a simplistic *more is better* policy would be misguided. A more nuanced approach might involve incentivizing an *optimal* level of diversity to mitigate the risk of decline and

encouraging cross-disciplinary collaboration, which could help shift the national research landscape from its current incremental focus toward more transformative, high-risk endeavors. Such policy changes need to be field-specific.

Based on a previous review outlining the future agenda [34], this study suggests further investigation into the underlying mechanisms of team dynamics (e.g., integrating disparate methods) in research groups with higher diversity. Furthermore, the intriguing result that groups on an Advancement trajectory share a similar diversity structure with Volatile groups calls for longitudinal research to track their long-term outcomes and identify the factors that lead to sustained success versus instability. Future quantitative work could also focus on modeling the non-linear relationship between diversity and prestige more precisely to identify potential *tipping points* where the benefits of diversity may be outweighed by coordination challenges.

## Supporting information

**S1 Table. Median DIV by group rank and size, 2013–2021.**
(DOCX)

## Author contributions

**Conceptualization:** Julian D. Cortes.

**Data curation:** Julian D. Cortes.

**Formal analysis:** Julian D. Cortes.

**Methodology:** Julian D. Cortes.

**Validation:** Julian D. Cortes.

**Visualization:** Julian D. Cortes.

**Writing – original draft:** Julian D. Cortes.

**Writing – review & editing:** Julian D. Cortes.

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
