## [Decision Letter · Decision Letter 0]

5 Dec 2025

Dear Dr. Cortes,

Thank you for submitting your manuscript to PLOS ONE. After careful consideration, we feel that it has merit but does not fully meet PLOS ONE’s publication criteria as it currently stands. Therefore, we invite you to submit a revised version of the manuscript that addresses the points raised during the review process.

We look forward to receiving your revised manuscript.

Kind regards,

Claudia Noemi González Brambila, Ph.D.

Academic Editor

PLOS One

Journal Requirements:

2. We notice that your supplementary table is included in the manuscript file. Please remove them and upload them with the file type 'Supporting Information'. Please ensure that each Supporting Information file has a legend listed in the manuscript after the references list.

Reviewers' comments:

Reviewer's Responses to Questions

**Comments to the Author**

1. Is the manuscript technically sound, and do the data support the conclusions?

Reviewer #1: Yes

Reviewer #2: Yes

Reviewer #3: Yes

2. Has the statistical analysis been performed appropriately and rigorously?

Reviewer #1: Yes

Reviewer #2: Yes

Reviewer #3: Yes

3. Have the authors made all data underlying the findings in their manuscript fully available?

Reviewer #1: Yes

Reviewer #2: Yes

Reviewer #3: Yes

4. Is the manuscript presented in an intelligible fashion and written in standard English?

Reviewer #1: Yes

Reviewer #2: Yes

Reviewer #3: Yes

Reviewer #1: Dear Author,

I have reviewed your manuscript and found it to be well-written and presenting a compelling contribution to the field. The content is likely to be of significant interest to readers and researchers in this area.

I recommend ensuring that all references cited are current and up-to-date. This will help to contextualize your findings within the latest developments in the field.

Congratulation

Reviewer #2: With abundant data and appropriate methods, the overall logic is clear. But the clarity of the charts in the article needs to be improved, for example, replacing Table 1 with a graphical form would be clearer. In addition, regarding the unclear description of the dataset on Page 8, it is suggested to add a table describing the dataset and provide annual data statistics

Reviewer #3: An identical manuscript is already published on Internet - Research Gate. (https://www.researchgate.net/publication/395028979_A_Little_Bit_of_This_A_Little_Bit_of_That_-_Disciplinary_Diversity_and_Scientific_Prestige_in_Research_Groups_of_Colombia). do now know it this is possible and it could be considered that ” Results reported have not been published elsewhere.” and important criteria of Plos publication.

**Do you want your identity to be public for this peer review?** For information about this choice, including consent withdrawal, please see our Privacy Policy

Reviewer #1: **Yes:** Ebenezer Ad Adams

Reviewer #2: No

Reviewer #3: No

---

## [Author Response · Author response to Decision Letter 1]

6 Dec 2025

Dear editor and reviewers, see attached the letter of response. Thanks.

---

## [Decision Letter · Decision Letter 1]

25 Jan 2026

Dear Dr. Cortes,

Thank you for submitting your manuscript to PLOS ONE. After careful consideration, we feel that it has merit but does not fully meet PLOS ONE’s publication criteria as it currently stands. Therefore, we invite you to submit a revised version of the manuscript that addresses the points raised during the review process.

We look forward to receiving your revised manuscript.

Kind regards,

Claudia Noemi González Brambila, Ph.D.

Academic Editor

PLOS One

Journal Requirements:

Reviewers' comments:

Reviewer's Responses to Questions

**Comments to the Author**

Reviewer #4: (No Response)

Reviewer #5: All comments have been addressed

2. Is the manuscript technically sound, and do the data support the conclusions?

Reviewer #4: Yes

Reviewer #5: Yes

3. Has the statistical analysis been performed appropriately and rigorously?

Reviewer #4: Yes

Reviewer #5: Yes

4. Have the authors made all data underlying the findings in their manuscript fully available?

Reviewer #4: Yes

Reviewer #5: Yes

5. Is the manuscript presented in an intelligible fashion and written in standard English?

Reviewer #4: Yes

Reviewer #5: Yes

Reviewer #2: Summary

With abundant data and appropriate methods, the overall logic is clear. But the clarity of the charts in the article needs to be improved, for example, replacing Table 1 with a graphical form would be clearer. In addition, regarding the unclear description of the dataset on Page 8, it is suggested to add a table describing the dataset and provide annual data statistics

Reviewer #4: Summary

This study examines the evolution of disciplinary diversity (DIV) in Colombian research groups and its relationship with scientific prestige, group size, rank, and performance trajectories over six evaluation periods using national administrative data. Overall, disciplinary diversity varies systematically by scientific area, group size, and rank, but its relationship with prestige and advancement is non-linear and weak in magnitude.

In most fields, groups on advancement and stagnation trajectories display similar median DIV, with the Social Sciences as a notable exception where stagnating groups show higher diversity. A consistent pattern across all areas is that groups in decline exhibit the lowest median diversity.

The findings suggest that disciplinary diversity should be considered in research evaluation and policy, but cautiously. Given the small effect size and non-monotonic relationship, policies promoting diversity should aim for field-specific “optimal” levels rather than indiscriminate increases. Future research should investigate the internal dynamics of diverse teams, explore long-term outcomes of volatile versus advancing groups, and model potential tipping points where the coordination costs of diversity may outweigh its benefits.

Overall, the manuscript is well written and follows a clear argumentative logic. The research questions are well motivated, and the methodological approach is appropriate. While the results are clearly presented, the discussion section could be strengthened by better contextualizing the findings and clarifying their broader implications.

General Comments

The discussion would benefit from a more explicit comparison between the effects of increasing group size and disciplinary diversity on scientific prestige. While both dimensions are analyzed, the manuscript could more clearly disentangle whether prestige gains are primarily driven by larger group size (for example through productivity, resource pooling, or visibility) rather than diversity per se, and how these two factors interact.

The discussion remains closely tied to the variables analyzed and could be broadened by situating the findings within the wider literature on research performance. In particular, factors beyond disciplinary composition—such as leadership quality, coordination capacity, institutional support, access to funding, international collaboration, and evaluation incentives—may influence research group rankings and could act as mediators or confounding factors.

Relatedly, the manuscript could benefit from a more explicit consideration of “soft” factors, such as communication, conflict management, leadership, and collaborative norms. These elements may have an impact that is orthogonal to, larger than, or even confounding with disciplinary diversity and group size. Discussing how such team-level dynamics might condition the effectiveness of diversity would add depth to the analysis.

While the focus on Colombia is a clear strength of the paper, the discussion could more explicitly address the extent to which the results can be generalized to other South American countries or to national research systems more broadly.

Detailed Comments

Abstract: Change “Leydesdorff et al.” to “Leydesdorff et al.’s”.

Abstract: Change “on an volatile” to “on a volatile”.

Introduction: In the sentence “Modern society and science face complex challenges,” please provide concrete examples of such challenges.

Introduction: In the sentence beginning with “This collaboration emerges…”, do you refer to scientific collaborations in general? Please clarify.

Introduction: Change “research in the field of the science of team science (SciTS) have examined” to “has examined”.

Related Literature (second-to-last paragraph): Please define the abbreviation STEM.

Methods 3.1 (Reference 44): The year in the R reference appears to correspond to the release date of the R version used. I would expect a newer version than 3.1 to have been used; please adjust accordingly.

Section 3.2.2 (Sampling): The subtitle “Sampling” is misleading, as no actual sampling procedure is described. I suggest changing it to “Selection” or a similar term.

Section 3.2.2, paragraph 3: The scientific ranks are introduced here, but their meaning and assessment are explained later in subsection 3.3.2. It would improve clarity to restructure these sections or to indicate explicitly that further details follow.

Tables and Figures: All table and figure captions should end with a period.

Figure 2: The y-axis labels are very long. Consider abbreviating them, for example as “# RGs”.

Equation (1): Please clarify what the subscript “d” refers to. If it indicates the group, it is unclear why N would also need a subscript.

Equation (2): The meaning of the symbol for the average value is unclear and should be specified.

Equations (3, 4, 6, 7): The notation for scalar multiplication is inconsistent. Please use a single convention throughout (for example, a multiplication dot or a multiplication sign).

Figure 3: Please explain the meaning of the different shapes used for Disparity. The color coding for Variety and Balance is intuitive, but the interpretation of Disparity is not clear.

Table 4: This table occupies a substantial amount of space, but its content could likely be summarized in a few sentences. Please consider removing it.

Figures 4, 5, and 6: These figures show the development over time of median DIV and related measures for different scientific areas and group sizes. However, the time axis is not evenly spaced, which visually overstates some effects. The time axis should be adjusted. In addition, there is a large amount of unused vertical space; adjusting the y-axis limits would improve readability.

Figure 5: Since there is only one median number of researchers per group rank, the legends could be combined. As currently shown, the legend suggests additional combinations that are not present. If only a subset is displayed, this should be clearly stated in the caption.

Figure 10: Since median DIV values are used for the subsequent statistical tests, it would be helpful to also visualize the corresponding variability (for example, standard deviations or interquartile ranges).

Reviewer #5: The manuscript “A Little Bit of This, A Little Bit of That — Disciplinary Diversity and Scientific Prestige in Research Groups of Colombia” ( PONE-D-25-47140R1 ) analyzes disciplinary diversity using extensive research output data across Colombia, as it relates to the national evaluation of research groups and investments in science. The main question addressed is whether team diversity is “inherently functional for team effectiveness”.

While this is a longstanding question, around which there are still unknowns and definitive evidence given the challenges of performing definitive social experiments on the matter – this study does approach it from a valuable perspective – in particular focusing on the ‘cognitive integration processes’ that are essential for team effectiveness, against the backdrop of national research system evaluation.

Overall the study is relatively descriptive in nature, being constructed upon extensive/official/appropriate data, applies appropriate statistical methods to provide a valuable framework for future studies, and visuals to convey the relative share of observations belonging to the various temporally disaggregated official categories defining disciplines, and the diversity observed therein, and the degree to which they correlate with other factors, such as team size. Consequently, the results are likely to be very useful within the case study region, as well as for other countries/regions faced with similar assessment questions and needs.

As such, I recommend publication as the author has addressed the comments in the prior reviews to the best of my ability to assess.

Main technical comment:

- I found it difficult to identify the definition of rank (A, A1, B, C) – which actually occurs in Section 3.3.2, well after they are mentioned in 3.2.2. These should be mad more explicitly clear up front, so that one does not need to infer that A1 > A > B > C; and also, to address why is there an A1 but no B2, etc; Also the first several Table/Plots would suggest that A>A1, which could be confusing.

Additional comments:

- I’m not sure what is the objective or value of the flippant title – I would recommend to remove the header “A Little Bit of This, A Little Bit of That”, as it is vague and does not contribute anything substantial or informative.

- Regarding the single-sentence statement - “In sum, evidence showed that, while diversity may offer benefits in specific, cognitively demanding contexts, it is not a universal performance enhancer” – here are two studies providing strong evidence in support of a causal link between team disciplinary diversity and broader citation impact:

Petersen AM, Majeti D, Kwon K, Ahmed ME, Pavlidis I. Cross-disciplinary evolution of the genomics revolution. Science advances. 2018 Aug 15;4(8):eaat4211.

Petersen AM, Ahmed ME, Pavlidis I. Grand challenges and emergent modes of convergence science. Humanities and Social Sciences Communications. 2021 Aug 4;8(1):1-5.

**Do you want your identity to be public for this peer review?** For information about this choice, including consent withdrawal, please see our Privacy Policy

Reviewer #4: No

Reviewer #5: No

---

## [Author Response · Author response to Decision Letter 2]

29 Jan 2026

Dear editor(s) and reviewers,

Thanks for your insights and recommendations to improve the manuscript. Please find attached in the file "Response to reviewers" the reviewers’ questions, our response, and the corresponding addition to the manuscript.

Sincerely,

The authors.

---

## [Decision Letter · Decision Letter 2]

11 Feb 2026

The Mosaic of Science — Disciplinary Diversity and Scientific Prestige in Research Groups in Colombia

PONE-D-25-47140R2

Dear Dr. Cortes,

We’re pleased to inform you that your manuscript has been judged scientifically suitable for publication and will be formally accepted for publication once it meets all outstanding technical requirements.

Kind regards,

Claudia Noemi González Brambila, Ph.D.

Academic Editor

PLOS One

Additional Editor Comments (optional):

Reviewers' comments:

Reviewer's Responses to Questions

**Comments to the Author**

Reviewer #4: All comments have been addressed

2. Is the manuscript technically sound, and do the data support the conclusions?

Reviewer #4: Yes

3. Has the statistical analysis been performed appropriately and rigorously?

Reviewer #4: Yes

4. Have the authors made all data underlying the findings in their manuscript fully available?

Reviewer #4: Yes

5. Is the manuscript presented in an intelligible fashion and written in standard English?

Reviewer #4: Yes

Reviewer #4: (No Response)

**Do you want your identity to be public for this peer review?** For information about this choice, including consent withdrawal, please see our Privacy Policy

Reviewer #4: No

---

## [Editor Report · Acceptance letter]

PONE-D-25-47140R2

PLOS One

Dear Dr. Cortes,

I'm pleased to inform you that your manuscript has been deemed suitable for publication in PLOS One. Congratulations! Your manuscript is now being handed over to our production team.

Kind regards,

on behalf of

Dr. Claudia Noemi González Brambila

Academic Editor

PLOS One